# Comparison of Perioperative Outcomes Using the da Vinci S, Si, X, and Xi Robotic Platforms for BABA Robotic Thyroidectomy

**DOI:** 10.3390/medicina57101130

**Published:** 2021-10-19

**Authors:** Hye Rim Shin, Keunchul Lee, Hyeong Won Yu, Su-jin Kim, Young Jun Chai, June Young Choi, Kyu Eun Lee

**Affiliations:** 1Department of Surgery, Seoul National University Bundang Hospital, Seongnam-si 13620, Korea; smallwheel91@gmail.com (H.R.S.); curitty@gmail.com (K.L.); 2Department of Surgery, Seoul National College of Medicine and University Hospital, Seoul 03080, Korea; su.jin.kim.md@gmail.com (S.-j.K.); kyueunlee@snu.ac.kr (K.E.L.); 3Boramae Medical Center, Department of Surgery, Seoul National University, Seoul 07061, Korea; kevinjoon@naver.com

**Keywords:** bilateral axillo-breast approach, BABA, robot, robotic surgery, comparison study

## Abstract

*Background and Objectives*: Robotic thyroidectomy via the bilateral axillo-breast approach (BABA), first introduced in Korea in 2008, has become a standard method of thyroid removal worldwide. The introduction of robotic surgical systems has enabled more patients to benefit from BABA robotic thyroidectomy, with good postoperative and excellent cosmetic results. To date, no studies have compared the benefits of the four currently available da Vinci robotic systems (S, Si, X, and Xi) for BABA robotic thyroidectomy. To determine the da Vinci model most suitable for BABA robotic thyroidectomy, the present study compared the perioperative outcomes in patients who underwent BABA robotic thyroidectomy using the four da Vinci models. *Materials and Methods*: This retrospective study evaluated outcomes in patients (*n* = 750) who underwent BABA robotic thyroidectomy using the four da Vinci systems from 2013 to 2019. The clinicopathologic data, including operation time, were compared. Substudy A compared the da Vinci models S and Si from 2013 to 2017, and substudy B compared models Si, X, and Xi from 2018 to 2019. *Results*: Substudy A, comparing the da Vinci S and Si systems, found no statistically significant differences between the two groups, whereas substudy B found that operation time was shorter in patients who underwent BABA robotic thyroidectomy with the da Vinci Xi system than with the Si and X systems. *Conclusions*: The da Vinci model Xi system can benefit patients undergoing BABA robotic thyroidectomy by shortening the operation time.

## 1. Introduction

Advances in robotic technology over the past 20 years has made robotic surgery commonplace. In 2000, the da Vinci robot (Intuitive Surgical, Inc., Sunnyvale, CA, USA) became the first robotic-assisted, minimally invasive surgical system to be approved for general laparoscopic surgery by the U.S. Food and Drug Administration (FDA) [1]. Several models of the da Vinci robot have since been introduced, including the S, Si, and X models, with the multiport Xi model introduced in 2014. The da Vinci robot was first utilized for mitral valve repair surgery in 1998 [2,3,4]. As of 2019, the da Vinci system has been utilized in more than 7 million procedures in 67 countries worldwide [5]. Hospitals may purchase different da Vinci robot models according to their preferences, with hospitals possessing and using more than one model at the same time. The presence of several robot versions within a hospital has enabled single-center studies comparing perioperative complications and disease prognosis following surgery using different da Vinci systems, with many of these studies assessing outcomes in departments of general surgery, otorhinolaryngology, urology, and gynecology. To date, however, few studies have compared differences among the various da Vinci systems in patients undergoing thyroidectomy.

Researchers and physicians may differ in their preference for robotic systems. For example, an analysis of patients undergoing colorectal surgery found that, compared with the Si platform, the new Xi platform improved the docking and operation time of total mesorectal excision for rectal cancer, but there were no differences between the Si and Xi in clinical and functional outcomes [6]. Moreover, perioperative short-term outcomes were similar in patients who underwent rectal cancer surgery using the Si and Xi systems, whereas console times were shorter with the Xi system [7]. By contrast, a study of patients who underwent sigmoidectomies and low anterior resection with the da Vinci Si and Xi platforms found no significant differences in operation time between these two systems [8].

Robot models have also been compared in patients undergoing gastric surgery for cancer. For example, outcomes were similar in patients who underwent surgery with the da Vinci Si and Xi systems [9]. By contrast, operation times were found to be shorter and patient prognosis enhanced following Xi-assisted Roux-En-Y gastric bypass [10]. Similarly, outcomes of robotic models have been compared in patients undergoing head and neck surgery. A comparison of the da Vinci Si and Xi systems for transoral robotic thyroidectomy showed that the Xi system resulted in better patient outcomes, including reductions in postoperative pain and hospital stay and fewer complications, although the surgical and console times were similar [11]. By contrast, overall operation and console times were significantly shorter in patients who underwent transoral robotic surgery for tongue cancer using the da Vinci Xi than the Si system [12].

The bilateral axillo-breast approach (BABA) method, involving access through the patient’s bilateral axillar and breast areolae, is frequently utilized for robotic thyroidectomy. Although several studies have reported the results of BABA robotic thyroidectomy results, none to our knowledge have compared the outcomes of da Vinci models. Therefore, the model most suitable for BABA robotic thyroidectomy remains unclear. The present study compared the sex, age, benign/malignant tumor status, extent of surgery, nodule size, pathology results, voice results, and operation time in patients who underwent robotic thyroidectomies at a single center using the da Vinci models S, Si, X, and Xi. In addition, this study evaluated whether any particular da Vinci model was optimal for BABA robotic thyroidectomy.

## 2. Materials and Methods

### 2.1. Study Design

This retrospective study analyzed patients who underwent BABA robotic thyroidectomy, performed by a single surgeon, from 28 March 2013, to 31 December 2019, at Seoul National University Bundang Hospital (SNUBH). The study protocol was approved by the hospital’s Institutional Review Board (IRB) (B-2101/661-102), which waived the requirement for informed consent due to the retrospective nature of this study. Four types of da Vinci robots have been introduced in SNUBH, S, Si, X, and Xi. The 6-year study period was divided into two analysis periods, depending on the use of each robot. Substudy A compared da Vinci models S and Si in patients who underwent thyroidectomy from 2013 to 2017, and substudy B compared da Vinci models Si, X, and Xi in patients who underwent thyroidectomy from 2018 to 2019 (Figure 1).

### 2.2. Patients

The study included patients with benign thyroid diseases, such as thyroid goiter and Graves’ disease, and malignant thyroid diseases, such as papillary thyroid carcinoma, follicular thyroid carcinoma, and medullary thyroid carcinoma. Patients’ medical records were reviewed to determine sex, age, benign/malignant tumor status, extent of surgery, nodule size, pathology results, voice results, and operation time. Only patients who underwent thyroid lobectomy were included; thus, patients who underwent total thyroidectomy, subtotal thyroidectomy, and complete thyroidectomy were excluded. Patients who underwent concomitant surgeries for other indications simultaneously with thyroid surgery and those who underwent modified radical neck dissection due to lateral lymph node (LN) metastasis were also excluded from the study. Total operation time was defined as the time from the start of the operation to the end of the operation, expressed in minutes. Vocal cord injury was defined as a loss of vocal cord function after surgery by direct laryngoscopy. Permanent hypoparathyroidism was defined as a postoperative parathyroid hormone concentration <15 pg/mL (normal range: ~15–65 pg/mL) for more than 1 year. Recovery within 1 year was defined as transient hypoparathyroidism.

### 2.3. BABA Robotic Thyroidectomy

The surgical procedures for BABA robotic thyroidectomy have been described [13]. BABA robotic thyroidectomy removes the thyroid gland using openings on both the axillary and breast areola areas without leaving a scar on the neck. In this method, a space of about 0.8–1.2 cm is made through four incisions, allowing the da Vinci robot to dock for surgery. The da Vinci S and Si models require a 1.2 cm wound for robot docking, whereas the da Vinci X and Xi models require a 0.8 cm opening, including a camera port. The surgical procedures involve (1) flap elevation, (2) midline division and isthmectomy, (3) lateral dissection, (4) inferior pole dissection, (5) preservation of the recurrent laryngeal nerve (RLN), (6) dissection of the Berry ligament, (7) superior pole dissection, (8) total thyroidectomy of the contralateral lobe, as above, and (9) specimen removal.

### 2.4. Comparisons of the da Vinci Models

All models of the da Vinci surgical system have three major components, the surgeon’s console, the patient-side cart, and the vision cart. The da Vinci S system, released in 2004, provides a highly magnified, three-dimensional high-definition (HD) view of the surgical area. It was designed according to a humanoid concept, with the EndoWrist instrument allowing a high degree of freedom of motion, enabling sophisticated work. The addition of a fourth manipulator arm and a slimmer arm design enabled greater access within the abdomen than that provided by the earlier version [14]. The camera and arm are connected with a separate 12 mm trocar. The da Vinci Si surgical system, released in 2009, is a full HD vision system, characterized by a dual-console mode with the ability to connect two operator consoles, allowing real-time teaching, surgical collaboration, and telementoring [15]. As with the da Vinci S surgical system, the camera and arm are connected with a separate 12 mm trocar.

The da Vinci Xi surgical system, released in 2014, has an easier docking system and a wider range of motion with smaller, thinner arms. The endoscope can be attached to any arm [7,16]. The overhead rotating arm suspension and anatomical targeting system provide better anatomical access for multi-quadrant operations [15,17]. An 8 mm trocar is used for docking all arms. The da Vinci X surgical system, released in 2017, is a hybrid version of the da Vinci Si and Xi surgical systems. It uses the same vision cart and surgeon console as the da Vinci Xi system, but the arms of the Xi system are installed on a da Vinci Si frame, resulting in an Xi surgeon console and vision cart and an Si patient cart [17]. The X system is less expensive than the da Vinci Xi system, while using the 8 mm trocar for docking.

### 2.5. Statistical Analysis

Clinicopathologic characteristics, surgical outcomes, surgical completeness, postoperative complications, and long-term results were compared in each substudy. To reduce any selection and time bias between groups, sex, age, benign/malignant tumor status, operation extent, nodule size, pathology results, voice results, and operation time were compared separately in substudies A and B.

Normally distributed continuous variables were expressed as means ± standard deviations and compared by Student’s *t*-tests, whereas non-normally distributed continuous variables were expressed as medians and compared by one-way analysis of variance (ANOVA). Categorical variables were expressed as numbers and percentages, and compared by Fisher’s exact test or Chi-squared test. All statistical analyses were performed using SPSS (version 22.0.0, IBM Corp., Armonk, NY, USA) for Windows, with *p* values <0.05 indicating statistical significance.

## 3. Results

Between March 2013 and December 2019, 1274 patients underwent robotic thyroidectomy via BABA at Seoul National University Bundang Hospital, with all operations performed by a single surgeon. Patients who underwent total thyroidectomy, subtotal thyroidectomy, and complete thyroidectomy were excluded (*n* = 482), as were patients who underwent lateral neck node dissection (*n* = 34) and concomitant surgery due to other diseases (*n* = 8).

Finally, the study population included 750 eligible patients (145 males and 605 females) of median age 40.0 years (range: 10–72 years). Their clinicopathologic characteristics are presented in Table 1. The mean malignant tumor size was 1.0 ± 0.8 cm (range: 0.1–6.5 cm) and the mean benign tumor size was 2.8 ± 1.3 cm (range: 0.8–5.5 cm). Postoperative pathologic examination showed that 693 patients had papillary thyroid carcinoma, 12 had follicular thyroid carcinoma, one had a medullary thyroid carcinoma, 16 had adenomatous goiters, and 28 had other tumors, including follicular adenoma, Hurthle cell adenoma, and Hurthle cell carcinoma. The median operating time for BABA thyroid lobectomy was 126.0 min (range: 73–245 min). Postoperatively, 15 (2.0%) patients had transient vocal cord injuries, but none had permanent vocal cord injuries.

Two substudies were performed to compare the da Vinci models used during the same period. Substudy A compared da Vinci models S and Si for 5 years (Table 2). Operative and postoperative variables, including tumor size, pathologic classification, and operation time, did not different significantly between these two models.

Substudy B compared da Vinci models Si, X, and Xi for 2 years (Table 3). Mean operation time was shortest in patients undergoing surgery with the Xi system (116.7 ± 22.9 min), followed by the Si system (124.7 ± 19.0 min), and the X system (127.5 ± 23.4 min), with operation time differing significantly among the three models (F = 7.157, *p* = 0.001). Bonferroni post hoc tests showed that the time required for thyroid surgery using the Xi system was significantly shorter than that using the X system (*p* = 0.001). However, the operation times required by the X and Xi systems did not differ significantly from that required by the Si system.

## 4. Discussion

Since its approval by the FDA in 2000, four types of multiport models of the da Vinci surgical system have become available, with each showing incremental improvements over the previous models [5]. Worldwide, this system was used in about 1,229,000 operations in 2019, an 18% increase compared with 2018 [18]. Robotic surgery has become a major part of minimally invasive surgery [19], with the increased worldwide use of robotic surgery accompanied by increases in research on these methods [20]. To date, however, most studies have compared robotic with open or endoscopic surgery, with few studies assessing the suitability of different robotic models or determining the optimal model for types of operation. Moreover, most previous studies have assessed the use of robotic surgery on diseases of the stomach and colon, and in urology and obstetrics and gynecology, with few studies assessing the use of robotic surgery on the thyroid gland [6,7,8,9,10,11,12].

BABA robotic thyroidectomy, first introduced in Korea in 2008, is frequently performed worldwide using da Vinci robotic systems [21]. To date, most studies of BABA robotic thyroidectomy have analyzed disease-related complication rates, with none comparing the suitability of different robotic models. The present study therefore compared outcomes of the da Vinci models to determine whether any of these was most suitable for BABA robotic thyroidectomy.

To match the study periods, robots used during the same time were compared, followed by cross-comparisons of the four models over the two study periods. Substudy A found no statistically significant differences between the da Vinci S and Si surgical systems, whereas substudy B found that the total operation time was shorter for the da Vinci Xi model than for the Si and X models. This result differed from that of a previous study, which found no significant differences between models in operating time on the thyroid gland [11].

Two reasons may explain the shorter operation time with the da Vinci Xi system. First, the operation time was shortened during the docking process. BABA robotic thyroidectomy is performed using both areolae and both axillae, with time required to connect the trocar to the robotic arm during the docking process. In the S and Si models, the robot body is connected to a plastic trocar, a difficult process that takes time for unskilled people. However, the integrated trocar provided in model Xi is changed into a form that can be easily coupled to the robot body, which may save time. Second, the trocar shaft can be lifted high after docking. An important preparatory process in BABA robotic thyroidectomy is to secure the field of view, requiring the robot arm to be slightly lifted after docking. The S and Si trocars are 1.2 cm thick, making them difficult to lift high due to the resistance of the skin and subcutaneous fat. However, Xi trocars are only 0.8 cm thick, allowing them to be lifted relatively smoothly to a high position. This difference can lead to a higher and wider field of view, allowing the operation to proceed more comfortably and smoothly. Despite our expectation, that the operation time with the da Vinci X model would also be reduced relative to the Si model, with the X and Xi models having similar operation times, no significant difference was observed between the X and Si models. This may have been due to an insufficient number of operations with the X model, reducing the statistical power of the comparison.

This study had several limitations, including its retrospective design. In addition, some of the groups that were compared differed slightly in age and sex ratios. Although propensity score matching may have provided more accurate results, in this case, this analysis was not performed due to concerns about excessively reducing the number of patients per group. Because this study was retrospective in design, console time and flap dissection time were not obtained and could not be analyzed. Although more patients underwent surgery using the Xi model than the other models, this limitation may have been overcome by the inclusion of more than 100 patients in each group. Prospective, large-scale studies are needed to compare these da Vinci robotic systems in thyroid surgery.

Some may argue that it is less important to compare previous and current models of robots. However, the model of the robot mainly used may be different depending on the country and place. We could use several robots in one institution. Based on these data, this study was conducted to confirm the differences between robots implemented in the same period.

## 5. Conclusions

This study found that there were no significant differences between da Vinci models S and Si in patients undergoing BABA robotic thyroidectomy. Importantly, use of the da Vinci model Xi could shorten BABA robotic thyroidectomy surgery time compared with the da Vinci models Si and X.

## Figures and Tables

**Figure 1 medicina-57-01130-f001:**
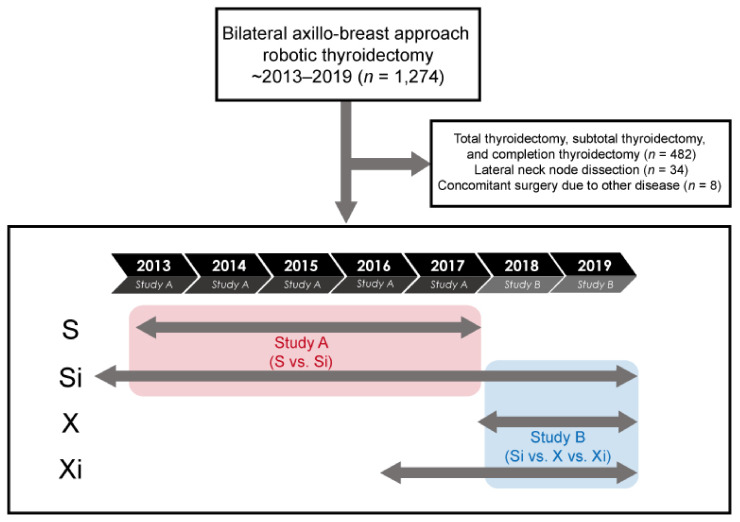
Study design.

**Table 1 medicina-57-01130-t001:** Clinicopathologic characteristics of the cohort.

	Total (*n* = 750)
Age (years)	40.0 (33.0–46.0)
Sex ratio (male:female)	1:4.2 (145:605)
Malignancy ratio (benign:malignant)	1:16.4 (43:707)
Extent of the surgery	
Lobectomy	750 (100%)
Mean tumor size (cm)	
Malignant	1.0 ± 0.8
Benign	2.9 ± 1.3
Pathologic classification	
Papillary thyroid carcinoma	693 (92.4%)
Follicular thyroid carcinoma	12 (1.6%)
Medullary thyroid carcinoma	1 (0.1%)
Adenomatous goiter	16 (2.1%)
Others	28 (3.7%)
Percentage of hypocalcemia	
Transient	12 (1.6%)
Permanent	1 (0.1%)
Percentage of vocal cord injury	
Transient	15 (2.0%)
Permanent	0 (0.0%)
Operation time (minutes)	126.0 (108.0–148.0)

**Table 2 medicina-57-01130-t002:** Substudy A comparing da Vinci models S and Si.

	S (*n* = 92)	Si (*n* = 92)	*p*
Age (years)	39.5 ± 9.3	38.9 ± 9.0	0.647
Sex ratio (male:female)	1:4.1 (18:74)	1:17.4 (5:87)	0.004
Malignancy ratio (benign:malignant)	1:22 (4:88)	1:17.4 (5:87)	0.733
Extent of the surgery			
Lobectomy	92 (100%)	92 (100%)	1.000
Mean tumor size (cm)			
Malignant	0.8 (0.6–1.1)	0.7 (0.5–0.9)	0.098
Benign	3.1 ± 0.8	3.3 ± 2.1	0.838
Pathologic classification			0.372
Papillary thyroid carcinoma	86 (93.5%)	86 (93.5%)	
Follicular thyroid carcinoma	2 (2.2%)	1 (1.1%)	
Medullary thyroid carcinoma	0 (0.0%)	0 (0.0%)	
Adenomatous goiter	1 (1.1%)	4 (4.3%)	
Others	3 (3.3%)	1 (1.1%)	
Percentage of hypocalcemia			0.097
Transient	5 (5.4%)	1 (1.1%)	
Permanent	0 (0.0%)	0 (0.0%)	
Percentage of vocal cord injury			0.155
Transient	0 (0.0%)	2 (2.2%)	
Permanent	0 (0.0%)	0 (0.0%)	
Operation time (mins)	153.5 (133.0–181.3)	158.5 (136.0–175.5)	0.972

**Table 3 medicina-57-01130-t003:** Substudy B comparing da Vinci models Si, X, and Xi.

	Si (*n* = 21)	X (*n* = 67)	Xi (*n* = 387)	F	*p*
Age (years)	42.9 ± 10.0	41.1 ± 9.1	40.5 ± 10.4	0.631	0.533
Sex ratio (male:female)	1:6 (3:18)	1:2.7 (18:49)	1:3.7 (83:304)		0.422
Malignancy ratio (benign:malignant)	1:20 (1:20)	1:10.2 (6:61)	1:18.4 (20:367)		0.458
Extent of the surgery					
Lobectomy	21 (100%)	67 (100%)	387 (100%)		1.000
Mean tumor size (cm)					
Malignant	0.9 ± 0.5	1.1 ± 0.6	1.1 ± 0.8	0.578	0.561
Benign	4.7	2.3 ± 1.4	2.8 ± 1.0	2.080	0.147
Pathologic classification					0.946
PTC	20 (95.2%)	60 (89.6%)	357 (92.2%)		
FTC	0 (0.0%)	1 (1.5%)	8 (2.1%)		
MTC	0 (0.0%)	0 (0.0%)	1 (0.3%)		
AG	0 (0.0%)	1 (1.5%)	6 (1.6%)		
Others	1 (4.8%)	5 (7.5%)	15 (3.9%)		
Percentage of hypocalcemia					0.536
Transient	0 (0.0%)	2 (3.0%)	3 (0.8%)		
Permanent	0 (0.0%)	0 (0.0%)	1 (0.3%)		
Percentage of vocal cord injury					0.063
Transient	1 (4.8%)	4 (6.0%)	6 (1.6%)		0.064
Permanent	0 (0.0%)	0 (0.0%)	0 (0.0%)		
Operation time (mins)	124.7 ± 19.0	127.5 ± 23.4	116.7 ± 22.9	7.157	0.001

PTC, papillary thyroid carcinoma; FTC, follicular thyroid carcinoma; MTC, medullary thyroid carcinoma; AG, adenomatous goiter.

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
