# Peer review of "Comparison of Perioperative Outcomes Using the da Vinci S, Si, X, and Xi Robotic Platforms for BABA Robotic Thyroidectomy"

_medicina, 2021, doi:10.3390/medicina57101130_

Round 1
Reviewer 1 Report
well structured article but with little scientific value
Author Response
Thank you for your opinion. Based on this study, we will plan for greater prospective research in the future to improve the quality of the study.
We attached the English revision certification as below.

Reviewer 2 Report
A nice comparison of results between different systems. It is rare to have one surgeon working across 4 different systems over the years and great that you were able to capture such data to provide such a comparison. I have attached some minor suggested edits in the attached document. One thing could me made more clear, namely your mentioning the difference between the trocars on the S, Si systems vs. the X, Xi systems. The camera port is definitely different (12 vs 8mm) and I assume you are using a 12mm port for a harmonic ace when using the S, Si system, accounting for the other 12mm port. What is used instead on the X, Xi system such that you don't need a 12mm trocar?

Author Response
Point 1: A nice comparison of results between different systems. It is rare to have one surgeon working across 4 different systems over the years and great that you were able to capture such data to provide such a comparison. I have attached some minor suggested edits in the attached document. One thing could me made more clear, namely your mentioning the difference between the trocars on the S, Si systems vs. the X, Xi systems. The camera port is definitely different (12 vs 8mm) and I assume you are using a 12mm port for a harmonic ace when using the S, Si system, accounting for the other 12mm port. What is used instead on the X, Xi system such that you don't need a 12mm trocar?
Response 1: Thank you for your opinions. I will answer each minor edition that you pointed out in the attached document.
Point 2: Question n page 2 of pdf file,
In general, do you think the shorter operative time on the Xi vs the Si are due to the ease of performing multi-quadrant surgery with the Xi system?
Response 2: As we mentioned in the discussion, the da Vinci Xi is characterized by ease of docking, with arms thinner than 1 cm and relatively easy to dock. Therefore, it is thought that Xi exhibits shorter operation time than Si, which is thicker than 1cm and is difficult to dock.
Point 3: Question in page 3 of pdf file,
For the camera only or are all of incisions 12mm?
Response 3: Thanks for your good question. We have changed the sentence as follows.
‘whereas the da Vinci X and Xi models require a 0.8 cm opening’
à ‘whereas the da Vinci X and Xi models require a 0.8 cm opening including camera port.’
Point 4: Question in page 7 of pdf file,
This is because of the use of the harmonic ace? otherwise 8mm instruments can be used...
Response 4: The reason we can easily lift the robot arm is because we use 8mm instruments. We have already written the content on line 243 (page 8) as shown below.
‘The S and Si trocars are 1.2 cm thick, making them difficult to lift high due to the resistance of the skin and subcutaneous fat. However, Xi trocars are only 0.8 cm thick, allowing them to be lifted relatively smoothly to a high position.’
Point 5: English edition
We attached the English revision certification as below.
Point 6: minor English editing
- page 1, line 46 : standard (deleted)
- page 2, line 56 : robot (deleted)
- page 2, line 68 : gastric (deleted)
- page 4, line 147 : smaller (changed)
- page 4, line 150 : all arms (changed)
- page 4, line 164 : medians (changed)

This manuscript is a resubmission of an earlier submission. The following is a list of the peer review reports and author responses from that submission.
Round 1
Reviewer 1 Report
the only significant difference is not clinically relevant (about 10 mins in 2 hours). The operation time should be defined in methods.
The largest bias that makes this study irrelevant is the timing of the different robots and the learning curve of the surgeon. Basically it is impossible to say if it is the surgeon (single) getting faster or if the robots are getting better.
Furthermore BABA is used for total thyroidectomy which are excluded. Maybe the authors were thinking ABBA?
English needs improvement.
there are several data problems and incongruences, such as number of nerve injuries.